# An external focus of attention enhances table tennis backhand stroke accuracy in low-skilled players

Tomasz Niźnikowski[1]*, Weronika Łuba-Arnista[2]*, Paweł Arnista[2], Jared M. Porter[3], Hubert Makaruk[1], Jerzy Sadowski[1], Andrzej Mastalerz[4], Ewelina Niźnikowska[5], Andrew Shaw[3]

1 Faculty of Physical Education and Health in Biała Podlaska, Józef Piłsudski University of Physical Education in Warsaw, Biała Podlaska, Poland, 2 Faculty of Health Sciences, Lomza State University of Applied Sciences, Lomza, Poland, 3 Department of Kinesiology, Recreation, and Sport Studies, The University of Tennessee, Knoxville, United States of America, 4 Faculty of Physical Education, Józef Piłsudski University of Physical Education in Warsaw, Warsaw, Poland, 5 Faculty of Health Sciences, John Paul II University of Applied Sciences in Biala Podlaska, Biała Podlaska, Poland

* tomasz.niznikowski@awf.edu.pl (TN); wluba@ansl.edu.pl (WŁA)

**Data Availability Statement:** All relevant data are within the paper and its Supporting Information files.

## Abstract

The aim of the study was to determine the impact of internal and external (proximal and distal) attentional focus on table tennis backhand stroke accuracy in low-skilled players. Fifty-one undergraduate physical education (PE) students were randomly assigned to 3 groups: Group G1 (IF) was instructed to focus on the hand holding the paddle, Group G2 (EFP) was instructed to focus on the ball, while Group G3 (EFD) was instructed to focus on targets marked on the tennis table. The experimental groups followed identical instructions except for the instruction about the focus of attention. Participants were asked to score as many points as possible by hitting the ball inside the three smallest targets marked on the tennis table. They were required to do so using a backhand stroke. The practice session consisted of 45 trials in three blocks of backhand (15 trials at each target). A special scoring system was used to determine the accuracy of the strokes. One of the most important findings from the current research was that groups with an external focus of attention revealed significant improvements in accuracy in the post-test, while the group with an internal focus of attention achieved low training effects. No significant difference was observed between G2 (EFP) and G3 (EFD) in the delayed retention test, which indicates that proximal and distal attentional focus had similar effects on table tennis backhand stroke accuracy in low-skilled players.

## Introduction

The effectiveness of motor learning has been analyzed from a number of different scientific perspectives, e.g., behavioural, socio-cognitive, neurophysiological or neurocomputational [1]. There are a lot of factors that affect motor skill acquisition, e.g., variable vs. constant practice,

**Funding:** The authors received no specific funding for this work.

**Competing interests:** The authors have declared that no competing interests exist.

blocked vs. random practice, self-controlled practice, simple vs. complex movement task, learner's level of advancement, external vs. internal attentional focus instructions, augmented feedback or observational learning [1, 2]. Providing instructions to a learner during motor learning is a common practice used by teachers or coaches. These instructions can be described as associative (i.e., focusing on bodily sensation) and dissociative (i.e., blocking out sensations resulting from physical effort), broad and narrow, or external (i.e., toward the effect of the movement) and internal (i.e., toward the body movement [1–4]. In the literature there is solid evidence that an external focus of attention enhances motor outcomes more substantially than an internal focus of attention. It has been shown in a variety of tasks such as a balance task [5], discus throwing performance [6], standing long jump [7], sprint performance [8], tennis skills technique [9] accuracy tasks in hitting golf balls [10, 11], basketball free throw shooting [12, 13], dart throwing [14, 15], frisbee flying-disk throwing [16] or archery shooting [17]. Wulf and Lewthwaite proposed the Optimizing Performance through Intrinsic Motivation and Attention for Learning (OPTIMAL) theory of motor learning [1]. They suggest that motivational and external attentional factors contribute to successful outcomes in motor learning by strengthening the coupling of performers' goals to their movement actions.

The effects of focus of attention in motor learning and performance are explained by the "constrained action hypothesis" [18, 19]. According to this theory, an internal focus of attention may interfere with automatic motor control processes. In contrast, an external attentional focus may lead to enhanced effectiveness and efficiency of movement by allowing automatic control. It is important to note that an external focus of attention may be directed close to the body (proximal) or farther away from the body (distal) [2]. Generally, the distal attentional focus is more easily distinguished from one's own body movements than the proximal focus of attention, which leads to increased performance improvement [20]. This is confirmed by studies conducted with novice performers in dart throwing [21] and standing long jump [22]. However, Singh and Wulf [23] claim that the optimal distance of external focus of attention depends on the level of expertise. Low-skilled practitioners benefit the most from a proximal external focus, whereas a distal focus of attention is more appropriate for high-skilled performers. This assumption is in line with Roberts and Lawrence's research results [24]. They showed that in an aiming task, using a proximal external focus of attention led to better outcomes.

In summary, it is unclear which type of external attentional focus (proximal or distal) is more beneficial in applied experiments with performers of different levels of expertise [4]. Thus, further research in this vein is recommended. In addition, Keller et al. [25] suggest that racquet sports may be better suited for inducing an external focus of attention during training due to the instrumentation with external targets. Unfortunately, insufficient research has been conducted on attentional focus in racquet sports (e.g., badminton, table tennis, squash, tennis, etc.). Therefore, the aim of this experiment was to determine the impact of internal and external (proximal and distal) attentional focus on table tennis backhand stroke accuracy in low-skilled players.

## Materials and methods

Fifty-one right-handed participants (n = 12 women, n = 39 men, age 22.9 ± 1.8 years, body mass 71.2 ± 9.7 kg, height 176 ± 8.5 cm) were recruited from undergraduate physical education (PE) courses. They enrolled in table tennis classes (thirty 45-minute sessions) as part of the university curriculum. The sample size in the current study was selected based on sample sizes in similar studies (n = 11 [11], n = 10 [16], n = 10 [6]). Power analysis of the current research using G*Power Version 3.1.9.4 [26] showed that with an estimated moderate effect size, it was determined that a minimum of twelve participants were required in each group (effect

size = 0.25, power = 0.80, p = 0.05), so the recruited sample of seventeen participants in each group was considered appropriate.

The participants were required to have no lower or upper extremity injury over the last 3 months, and no previous organized table tennis training. Each participant was considered a low-skilled player who had basic understanding and skills of table tennis strokes, and all were naive to the study purpose. The participants were randomly assigned to one of the three practice groups: G1 (IF)–with a focus of attention on the hand holding the paddle (n = 17; age 22.4 ± 1.4 years, body mass 71.9 ± 10.4 kg, body height 175 ± 8.3 cm), G2 (EFP)–with a focus of attention on the ball (n = 17, age 23.1 ± 1.7 years, body mass 72.4 ± 10.5 kg, body height 175.8 ± 9.2 cm) and G3 (EFD)–with a focus of attention on targets marked on the tennis table (n = 17, age 23.2 ± 2.2 years, body mass 69.4 ± 8.4 kg, body height 177.1 ± 8.2 cm). Each group was equal in terms of the gender of the participants. All participants delivered written informed consent prior to the commencement of the experiment. Ethical approval was provided by the Senate Scientific Research Ethics Committee of Lomza State University of Applied Sciences (document code: 4175500, 22.11.2021).

The participants performed a pre-test, a practice phase and a post-test on the same day. The retention test was conducted 24 hours after the practice phase. During the tests and the practice phase, the participants were instructed to score as many points as possible by hitting the ball inside the three smallest targets marked on the tennis table (targets 1, 2, 3) (Fig 1).

The practice session consisted of 45 trials in three blocks (15 trials at each target). Also, during each test all the participants completed 45 trials in three blocks. The participants were required to perform backhand strokes. The experimental groups followed a similar experimental design, with one difference–instructions concerning the focus of attention were provided in the practice phase only (no attentional focus instructions were given in the pre-test or post-test). Participants from group G1 (IF) were instructed to "concentrate on the hand holding the paddle", group G2 (EFP) were told to "concentrate on the ball", while group G3 (EFD) were asked to "concentrate on targets marked on the tennis table". The experimenter gave attentional focus reminders at the beginning of each block. On each test day, the participants performed a 15-minute warm-up routine (running and flexibility exercises). A 30-second break was provided after each block. The post-test was administered immediately after the practice session. No attentional focus instructions or reminders were given in the retention test to assess relatively permanent effects of the instructions.

The experiment was conducted on consecutive days in a sports hall and was performed on a standard-size tennis table approved by the International Table Tennis Federation (ANDRO Magnum SC, Germany). Each of the participants used the same professional paddle with the following characteristics: blade (ANDRO "Inizio ALL", Germany) and rubbers (DONIC "Liga", 2.0 mm, Germany). Plastic table tennis balls (DONIC "Coach ** P40+ Cell-Free", Germany) were delivered to the participants by a table tennis robot (NEWGY Robo-Pong 1050, DONIC, Newgy Industries Inc., Tennessee, USA) positioned in the middle of the table, on the opposite side of the participant. The robot was programmed as follows: ball speed (level 13), frequency–the time between balls served (1.5 s; 40 balls/min), rotation type (topspin), the angle of the robot head (level 6), the ball placement on the table (for backhand strokes–left and right position: level 3). The assessment developed by Poolton et al. [27] was used to determine the accuracy of the strokes during the pre-test, post-test and retention test (Fig 1).

On the table opposite the participant, there were six squares (50 cm per side) and three smaller squares (25 cm per side) centrally located inside the furthest squares (targets 4, 5, 6) (Fig 1). Three points were awarded when the ball landed on the smallest targets (targets 1, 2, 3), two points were awarded when the ball hit into the large distal squares surrounding the smallest targets, and one point was awarded if the ball hit inside the three large proximal

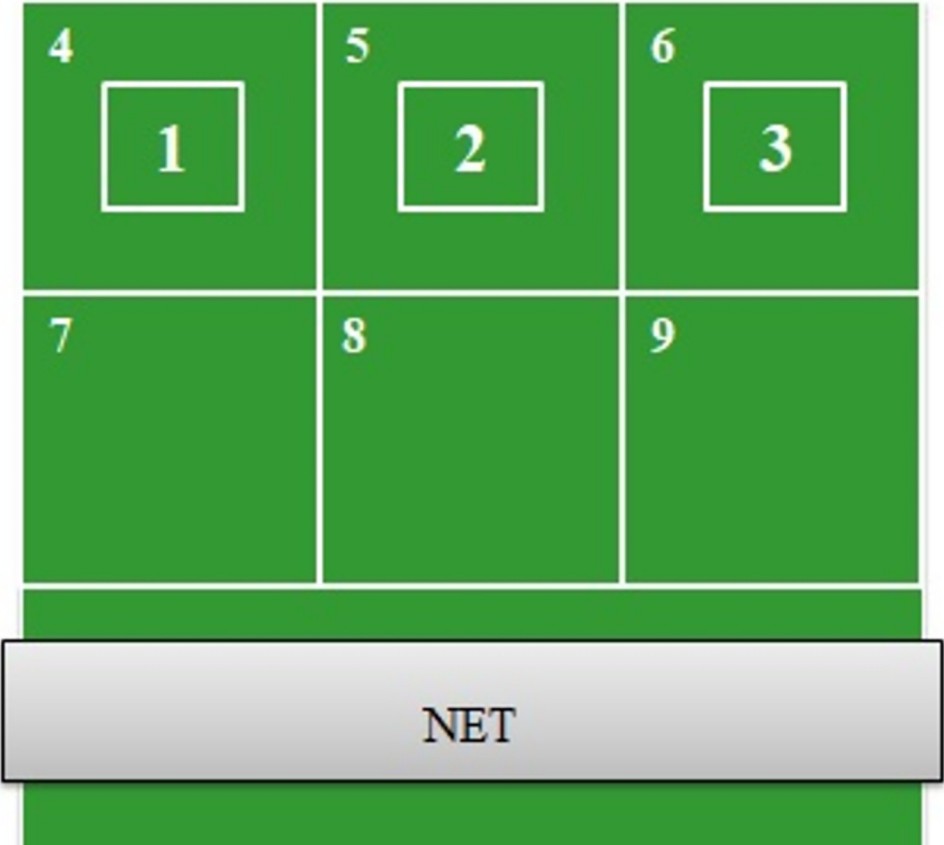

**Fig 1. Target location on the tennis table (number: 1, 2, 3).**

squares. The participants received no points if the ball did not hit any of the squares. In total, they performed 45 trials (15 trials per one target). Shots that landed in target number 3 scored 3 points, those in number 6 = 2 points, and those in number 9 = 1 point, while balls landing in all other squares scored 0 points. The points were summed across all strokes to determine the final performance score. The reliability of this test was confirmed by the value of the intra-class correlation coefficient (ICC = 0.92; excellent agreement) [28]. A single player with 17 years of table tennis experience recorded the points scored for all tests of every participant. No feedback about errors was provided to the participants other than the knowledge of results available to them by performing the task.

Prior to conducting the analysis, all data was checked to ensure it satisfied parametric assumptions of normality and homogeneity of variance using the Kolmogorov-Smirnov and Shapiro-Wilk tests. All violations of sphericity were corrected by adjusting the degree of freedom using the Greenhouse-Geisser correction when the sphericity estimate was less than 0.75, and the Huynh-Feldt correction when it was greater than 0.75 [29]. Effect sizes were reported using partial eta squared (np2) for all main effects and interactions and Cohen's d for comparison between two means. Cohen's d effect sizes were calculated using the following formula: Cohen's d = M1 –M2/spooled, where spooled = ($\sqrt{}$S1+S2)/2. Parametric effect sizes were defined as large d > 0.8, moderate 0.8 > d > 0.5 and small d < 0.5 [30]. A reported measures ANOVA was conducted to verify that there were differences in follow-through mean number of points (MNP) depending on the focus of attention on the hand holding the paddle, focus of

attention on the ball or focus of attention on targets marked on the tennis table. A further one-way between-participants ANOVA was run on pre-test accuracy scores to ensure there were no differences in performance levels between groups prior to the intervention. A 3 group x 3 period [pre-test, post-test (cumulates/effects), delayed training effects (retention test)] mixed design ANOVA with repeated measures on the last factor was employed to compare differences in MNP. Post-hoc Fisher's LSD was used to analyze significant main effects and interactions to determine the location of differences within (period) and between (group) factors

## Results

All groups performed with similar accuracy (MSP) on the pre-test (Table 1). The analysis of MNP did not reveal a main effect of Group ($F_{(2.48)} = 1.249$, $p = 0.296$, $\eta 2 = 0.049$). Training period ($F_{(2.96)} = 50.116$, $p < 0.001$, $\eta 2 = 0.511$) and the analysis of the interaction of those two factors ($F_{(4.96)} = 3.354$, $p = 0.0129$, $\eta 2 = 0.123$) reached statistical significance.

### Training effect

Post-hoc analyses revealed significant improvements in MNP from pre-test to post-test (Table 1) for group G2 (17.7%, $d = 0.37$, $p = 0.001$) and for group G3 (27.3%, $d = 0.51$, $p < 0.001$), while an insignificant increase in MNP was found for group G1 (5.5%, $d = 0.15$, $p = 0.262$). There were no differences in MNP between G1 and G2 ($d = 0.21$, $p = 0.699$). The mean number of points in G3 were higher than those in G1 ($d = 0.27$, $p = 0.047$).

### Delayed training effects/DET/

The analysis of DET revealed significant improvements in MNP over post-test (Table 1) for G1 (9.3%, $d = 0.24$, $p = 0.002$), G2 (13.7%, $d = 0.31$, $p < 0.001$) and G3 (10.6%, $d = 0.29$, $p < 0.001$). Post-hoc analysis indicated that DET results in group G3 were significantly higher compared to G1 ($d = 0.34$, $p = 0.017$). There was no significant difference between G3 and G2 ($d = 0.17$, $p = 0.206$).

## Discussion

The aim of the study was to determine the impact of internal and external (proximal and distal) attentional focus on table tennis backhand stroke accuracy in low-skilled players. Throughout the literature, it has been widely confirmed that an external focus of attention enhances motor outcomes more substantially than an internal focus of attention [2, 4].

**Table 1. Mean number of points (MNP) in G1, G2 and G3 during the two phases of the experimental design: Pre-acquisition and post-acquisition.**

| Group | Training period | Mean | SD |
|---|---|---|---|
| G1 (IF) | pre-test | 71.26 | 8.08 |
| | post-test | 75.19 | 10.28 |
| | retention test | 82.22 | 14.51 |
| G2 (EFP) | pre-test | 65.41 | 14.30 |
| | post-test | 76.99 | 13.13 |
| | retention test | 87.54 | 15.03 |
| G3 (EFD) | pre-test | 66.36 | 13.09 |
| | post-test | 84.46 | 19.20 |
| | retention test | 93.43 | 10.57 |

One of the most important findings from the current research was that groups with an external focus of attention (G2 and G3) revealed significant improvements in accuracy in the post-test, while the group with an internal focus of attention (G1) achieved a low short-term performance effect. This result is in line with the "constrained action hypothesis" [18, 19], which says that an internal focus of attention leads to conscious motor control which impairs automaticity and reduces movement efficiency [2]. In addition, some scientific evidence reported improved neuromuscular efficiency during an external focus of attention compared to internal focus in such tasks as isokinetic elbow flexions [31], isometric force production tasks [32] or isometric plantar-flexion [33]. Also, it was proved that an external focus of attention enhanced muscular endurance compared with an internal focus [34]. Hatami et al. [35] found that in no instruction situations, shoulder muscle activity did not change significantly as compared with the external focus of attention situation during forehand stroke in table tennis.

Our results showed that external attentional focus instructions accelerate the learning of the motor task so that the learner can achieve a higher skill level sooner [36]. However in the post-test, no significant differences between groups G2 and G1 were observed. Still, significant differences were noted between G3 and G1. Performance at the immediate assessment phase is often not predictive of outcomes at the time of a delayed retention test due to the process of motor memory consolidation [37]. Admittedly, each of the groups significantly improved the accuracy in the delayed retention test over the post-test. It was also noted that the group with a focus of attention on targets (G3) achieved a significant higher outcome than the group with a focus of attention on the hand holding the paddle (G1). This result is consistent with the findings of Chua et al. [38] that a distal external attentional focus facilitates both immediate performance and longer-term learning.

The lack of significant differences between G2 and G3 in the delayed retention test may indicate that both proximal and distal attentional focus is similarly effective in table tennis backhand stroke accuracy in low-skilled players. This finding is contrary to earlier research results that compared distal and proximal attentional focus in motor performance [21–24]. McNevin et al. [20] claim that increasing the distance of the external attentional focus enhanced motor learning. This "distance effect" has also been confirmed in other studies [7, 39, 40]. However, Singh and Wulf [23] claim that in the context of the "distance" of the external attentional focus, it is very important to consider the performer's skill level. Wulf and Prinz [41] suggest that performers without experience in a specific task might benefit more from a proximal attentional focus because they need to learn basic coordination. In turn, experienced performers might benefit from a distal attentional focus because this situation might facilitate the development of the movement pattern that is necessary to achieve the goal. It may explain the lack of significant differences between G2 and G3 in the delayed retention test. Possibly, the fact that low-skilled performers who had a higher level of physical fitness than less active individuals of the same age were involved in this study could have had an impact on our study result. In addition, our study findings can plausibly be explained in that the focus of attention on the ball that was in motion all the time could be more difficult than the focus of attention on constant targets marked on the tennis table. In practice, PE teachers and coaches working with adolescents can support autonomy by allowing self-definition of success using an external cue to enhance effective goal-action coupling [1].

The limitation of the study is that researchers, coaches and practitioners cannot generalize the results from this analysis of one specific stroke in table tennis to the whole sports discipline. Thus, future studies are recommended to analyze the impact of internal and external (proximal and distal) attentional focus while using other strokes in table tennis. What is more, Razaghi et al. [42] suggest that the combinations of external focus instructions and self-controlled feedback should be used to improve performance and motor learning outcomes. Also,

Makaruk et al. [43] propose that a combination of external attentional focus with autonomy support may produce benefits in motor performance. This line of research is worth further investigation in the area of table tennis.

## Conclusions

The findings of this study indicate that proximal and distal attentional focus is similarly effective in table tennis backhand stroke accuracy in low-skilled players' development. Also, these results suggest that table tennis coaches and practitioners can creatively use an external attentional focus to facilitate the performance of motor skills. To date, knowledge about the most effective external attentional focus (proximal or distal) in the performance of complex motor skills by practitioners at different skill levels is incomplete. Further research in this vein is recommended.

The funders had no role in study design, data collection and analysis, decision to publish, or preparation of the manuscript.

## Supporting information

**S1 File.**
(XLSX)

## Author Contributions

**Conceptualization:** Tomasz Niźnikowski, Weronika Łuba-Arnista, Jared M. Porter, Hubert Makaruk, Jerzy Sadowski, Andrzej Mastalerz, Ewelina Niźnikowska, Andrew Shaw.

**Data curation:** Tomasz Niźnikowski, Weronika Łuba-Arnista, Paweł Arnista, Hubert Makaruk, Jerzy Sadowski, Andrzej Mastalerz, Ewelina Niźnikowska.

**Formal analysis:** Tomasz Niźnikowski, Weronika Łuba-Arnista, Paweł Arnista, Jared M. Porter, Hubert Makaruk, Jerzy Sadowski, Andrzej Mastalerz, Ewelina Niźnikowska.

**Funding acquisition:** Tomasz Niźnikowski.

**Investigation:** Tomasz Niźnikowski, Weronika Łuba-Arnista, Paweł Arnista, Jerzy Sadowski.

**Methodology:** Tomasz Niźnikowski, Weronika Łuba-Arnista, Paweł Arnista, Jared M. Porter, Hubert Makaruk, Jerzy Sadowski, Andrzej Mastalerz, Ewelina Niźnikowska.

**Project administration:** Tomasz Niźnikowski, Weronika Łuba-Arnista, Paweł Arnista, Jared M. Porter, Hubert Makaruk, Jerzy Sadowski, Andrzej Mastalerz.

**Resources:** Tomasz Niźnikowski, Weronika Łuba-Arnista, Paweł Arnista, Andrzej Mastalerz.

**Software:** Weronika Łuba-Arnista, Paweł Arnista.

**Supervision:** Tomasz Niźnikowski, Jared M. Porter, Hubert Makaruk, Jerzy Sadowski, Andrzej Mastalerz.

**Validation:** Tomasz Niźnikowski, Weronika Łuba-Arnista, Paweł Arnista, Jared M. Porter, Hubert Makaruk, Jerzy Sadowski, Andrzej Mastalerz, Andrew Shaw.

**Visualization:** Tomasz Niźnikowski, Paweł Arnista, Jared M. Porter, Hubert Makaruk, Jerzy Sadowski, Andrzej Mastalerz, Ewelina Niźnikowska, Andrew Shaw.

**Writing – original draft:** Tomasz Niźnikowski, Weronika Łuba-Arnista, Paweł Arnista, Jared M. Porter, Hubert Makaruk, Jerzy Sadowski, Andrzej Mastalerz, Ewelina Niźnikowska, Andrew Shaw.

**Writing – review & editing:** Tomasz Niźnikowski, Weronika Łuba-Arnista, Paweł Arnista, Jared M. Porter, Hubert Makaruk, Jerzy Sadowski, Andrzej Mastalerz, Ewelina Niźnikowska, Andrew Shaw.

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
