## [Decision Letter · Decision Letter 0]

27 Sep 2022

PONE-D-22-24485An external focus of attention enhances table tennis backhand stroke accuracy in low-skilled playersPLOS ONE

Dear Dr. Tomasz Niznikowski, 

Thank you for submitting your manuscript to PLOS ONE. After careful consideration, we feel that it has merit but does not fully meet PLOS ONE’s publication criteria as it currently stands. Therefore, we invite you to submit a revised version of the manuscript that addresses the points raised during the review process.

ACADEMIC EDITOR:Dear authors, Although one of the reviewers considers your article suitable for publication, other for asked major revisions especially directed towards improving the introduction section (considering novel optimal theory on the focus of attention) and clarification of the methodology. 

We look forward to receiving your revised manuscript.

Kind regards,

Danica Janicijevic, Ph.D

Academic Editor

PLOS ONE

Journal Requirements:

"NO - Include this sentence at the end of your statement: The funders had no role in study design, data collection and analysis, decision to publish, or preparation of the manuscript."

Reviewers' comments:

Reviewer's Responses to Questions

**Comments to the Author**

1. Is the manuscript technically sound, and do the data support the conclusions?

Reviewer #1: Partly

Reviewer #2: Yes

2. Has the statistical analysis been performed appropriately and rigorously? 

Reviewer #1: N/A

Reviewer #2: Yes

3. Have the authors made all data underlying the findings in their manuscript fully available?

Reviewer #1: Yes

Reviewer #2: Yes

4. Is the manuscript presented in an intelligible fashion and written in standard English?

Reviewer #1: Yes

Reviewer #2: Yes

5. Review Comments to the Author

Reviewer #1: 1) The text needs to be improved grammatically.

Introduction

2) In the beginning of the Introduction, you are needed to add a brief introduction of various teaching strategies in motor learning. Then you need to enter the attentional focus as an important strategy. And then, please focus on different types of attentional focus (i.e., internal vs. external).

3) Introduction is too weak. The most recent theory on focus of attention is the OTPIAML theory. The authors have not explained this theory in the Introduction. In addition, the novelty of this study is not clear. If using a racquet sport enough for novelty of this experiment. What is the difference between racquet and not-racquet sports? These issues need to be explained more clearly in the Introduction.

Method

4) Why you chose low-skilled players and not absolutely novices?

5) How the students were low-skilled? Were they in the beginnings of semester, mid of semester, or have already finished they courses?

6) Is the info about BMI and etc. important?

7) The info about sample size should be placed in the Participants.

8) The most important issue in your study in attentional focus instructions. However, it is not clear how these instructions were provided and applied. The procedure of this study has not been included. How the participant performed the strokes? If a person from opposite side throws the ball? Or the participant took the ball in his/her hand? Please explain the procedure with all detail.

9) In relation to the comment “8”, it should be said that table tennis is considered as very fast sport, because the acceleration of the ball is very high. My most concern is that how the participates in G2 could concentrate on a ball which moves very fast? I am really confused with choosing this sport for examine proximal and distal effects. It is hard to believe that the participants could focus only on distal distance in table tennis, as the table tennis players focus mostly on balls not on the table. May shooting a basketball could provide more accurate findings.

Results

10) The table 1 looks very simply. Please use different table design.

11) You talked about so many statistical tests; however, they were not reported in the Results?

Discussion

12) It was assumed that the discussion held heavily on the differences between proximal and distal effects in attentional focus. However, based on the Discussion, it is not fully understood why proximal and distal effects are same in this study. Please clarify your findings.

13) Finally, an important reason for the effectiveness of external focus of attention is “goal-action coupling”. However, you have not pointed it in your discussion. Please use it as well as the OPTIMAL theory for discussion your results.

Reviewer #2: This is an appropriate paper I have no hesitation in recommending it. The statistics used are appropriate and the conclusion is consistent and sound. While not entirely novel, it could deserve dissemination across the scientific community. I am looking forward to your future success.

6. PLOS authors have the option to publish the peer review history of their article (what does this mean?). If published, this will include your full peer review and any attached files.

Reviewer #1: **Yes: **Saeed Ghorbani

Reviewer #2: **Yes: **Rei Odagiri

---

## [Author Response · Author response to Decision Letter 0]

2 Nov 2022

RESPONSE TO REVIEWERS COMMENTS 

Dear Editor and Reviewers, 

The authors would like to thank the reviewers for their precious time and invaluable comments. We have carefully addressed all the comments. The corresponding changes and refinements made in the revised paper are summarized in our response below. We hope that the modifications and explanations will be acceptable to you. 

REVIEWER #1 

1. Is the manuscript technically sound, and do the data support the conclusions?

Reviewer #1: Partly 

Response 1: We agree with the reviewer and have made corrections. 

Reviewer #2: Yes 

Response 1: Thank you for your opinion 

2. Has the statistical analysis been performed appropriately and rigorously? 

Reviewer #1: N/A 

Response 2: Thank you for your opinion 

Reviewer #2: Yes 

Response 2: Thank you for your opinion 

3. Have the authors made all data underlying the findings in their manuscript fully available?

The PLOS Data policy requires authors to make all data underlying the findings described in their manuscript fully available without restriction, with rare exception (please refer to the Data Availability Statement in the manuscript PDF file). The data should be provided as part of the manuscript or its supporting information, or deposited in a public repository. For example, in addition to summary statistics, the data points behind means, medians and variance measures should be available. If there are restrictions on publicly sharing data—e.g. participant privacy or use of data from a third party—those must be specified. 

Reviewer #1: Yes 

Response 3: Thank you for your opinion 

Reviewer #2: Yes 

Response 3: Thank you for your opinion 

4. Is the manuscript presented in an intelligible fashion and written in standard English?

Reviewer #1: Yes 

Response 4: Thank you for your opinion 

Reviewer #2: Yes 

Response 4: Thank you for your opinion 

5. Review Comments to the Author

Reviewer #1: 1) The text needs to be improved grammatically. 

Response 5.1: According to reviewers’ suggestions, manuscript has been reviewed and edited by a university native English speaker. 

Introduction

2) At the beginning of the Introduction, you are needed to add a brief introduction of various teaching strategies in motor learning. Then you need to enter the attentional focus as an important strategy. And then, please focus on different types of attentional focus (i.e., internal vs. external). 

Response 5.2: We agree with the reviewer and have made corrections. 

3) Introduction is too weak. The most recent theory on focus of attention is the OTPIAML theory. The authors have not explained this theory in the Introduction. In addition, the novelty of this study is not clear. If using a racquet sport is enough for novelty of this experiment. What is the difference between racquet and not-racquet sports? These issues need to be explained more clearly in the Introduction. 

Response 5.3: We agree with the reviewer and we change this part of the introduction. 

4) Why you chose low-skilled players and not absolutely novices? 

Response 5.4: Thank you for your question. We chose low-skilled players after 30 hours of table tennis classes because we wanted to verify how to conduct skill improvement. 

5) How the students were low-skilled? Were they at the beginning of the semester, mid of semester, or have already finished their courses? 

Response 5.5: Thank you for your question. The students were at the finished of the semester after 30 hours of table tennis classes. 

6) Is the info about BMI and etc. important? 

Response 5.5: Thank you for your question. BMI and etc. are very important in a research investigation, but this issue was not the purpose of our work. 

7) The info about sample size should be placed in the Participants. 

Response 5.7: We agree with the reviewer and we change this part of the methods. 

8) The most important issue in your study is attentional focus instructions. However, it is not clear how these instructions were provided and applied. The procedure of this study has not been included. How the participant performed the strokes? If a person from opposite side throws the ball? Or the participant took the ball in his/her hand? Please explain the procedure in all detail. 

Response 5.8: We respectfully disagree with the reviewer. The procedure of this study has been included. The experimental groups followed a similar experimental design, with one difference – instruction concerning focus of attention only in the practice phase (no attentional focus instructions were given in pre-test or post-test). Participants of G1 group (IF) were instructed to “concentrate on the hand holding the paddle”, G2 group (EFP) received instruction “concentrate on the ball”, and G3 group (EFD) – “concentrate on targets marked on the tennis table”. The experimenter gave attentional focus reminders at the beginning of each block. On each test day, the participants performed 15-minute warm-up routine (running and physical exercises). A 30-second break was provided after each block. The post-test was administered immediately after the practice session. No attentional focus instructions or reminders were given in retention test to assess the relatively permanent effects of the instructions. The experiment was conducted on consecutive days in a sports hall and was performed on a standard-size tennis table approved by International Table Tennis Federation (ANDRO Magnum SC, Germany). Each of the participants used the same professional paddle with the following characteristics: blade (ANDRO "Inizio ALL", Germany) and rubbers (DONIC "Liga", 2.0 mm, Germany). Plastic table tennis balls (DONIC "Coach ** P40+ Cell-Free", Germany) were delivered to the participants by a table tennis robot (NEWGY Robo-Pong 1050, DONIC, Newgy Industries Inc., Tennessee, USA) which was positioned in the middle of the table, on the opposite side of the participant. The robot was programmed as follows: ball speed (level 13), frequency – the time between balls served (1.5 s; 40 balls/min), rotation type (topspin), the angle of the robot head (level 6), the ball placement on the table (for backhand strokes – left and right position: level 3). The assessment developed by Poolton et al. [26] was used to determine the accuracy of the strokes during the pre-test, post-test and retention tests. 

9) In relation to the comment “8”, it should be said that table tennis is considered a very fast sport because the acceleration of the ball is very high. My most concern is how the participants in G2 could concentrate on a ball which moves very fast. I am really confused about choosing this sport to examine proximal and distal effects. It is hard to believe that the participants could focus only on distal distance in table tennis, as table tennis players focus mostly on balls, not on the table. May shooting a basketball could provide more accurate findings. 

Response 9: We agree with the reviewer that table tennis is difficult and complex sport. Therefore the robot was programmed as follows: ball speed (level 13), frequency – the time between of balls served (1.5 s; 40 balls/min), rotation type (topspin), the angle of the robot head (level 6), the ball placement on the table (for backhand strokes – left and right position: level 3). 

Results

10) The table 1 looks very simple. Please use a different table design. 

Response 10: We agree with the reviewer and we change this part of the results. 

11) You talked about so many statistical tests; however, they were not reported in the Results? 

Response 11: We agree with the reviewer and we change these parts. 

Discussion

12) It was assumed that the discussion held heavily on the differences between proximal and distal effects in attentional focus. However, based on the Discussion, it is not fully understood why proximal and distal effects are the same in this study. Please clarify your findings. 

Response 12: We agree with the reviewer and we change this part of the discussion. 

13) Finally, an important reason for the effectiveness of external focus of attention is “goal-action coupling”. However, you have not pointed it out in your discussion. Please use it as well as the OPTIMAL theory for discussing your results. 

Response 13: We agree with the reviewer and we change this part of the discussion. 

Reviewer #2: This is an appropriate paper I have no hesitation in recommending it. The statistics used are appropriate and the conclusion is consistent and sound. While not entirely novel, it could deserve dissemination across the scientific community. I am looking forward to your future success. 

Response 5: Thank you for your opinion

---

## [Decision Letter · Decision Letter 1]

18 Nov 2022

An external focus of attention enhances table tennis backhand stroke accuracy in low-skilled players

PONE-D-22-24485R1

Dear Dr. Tomasz Niznikowski,

We’re pleased to inform you that your manuscript has been judged scientifically suitable for publication and will be formally accepted for publication once it meets all outstanding technical requirements.

Kind regards,

Danica Janicijevic, Ph.D

Academic Editor

PLOS ONE

Additional Editor Comments (optional):

Reviewers' comments:

Reviewer's Responses to Questions

**Comments to the Author**

1. If the authors have adequately addressed your comments raised in a previous round of review and you feel that this manuscript is now acceptable for publication, you may indicate that here to bypass the “Comments to the Author” section, enter your conflict of interest statement in the “Confidential to Editor” section, and submit your "Accept" recommendation.

Reviewer #1: All comments have been addressed

2. Is the manuscript technically sound, and do the data support the conclusions?

Reviewer #1: Yes

3. Has the statistical analysis been performed appropriately and rigorously? 

Reviewer #1: Yes

4. Have the authors made all data underlying the findings in their manuscript fully available?

Reviewer #1: Yes

5. Is the manuscript presented in an intelligible fashion and written in standard English?

Reviewer #1: Yes

6. Review Comments to the Author

Reviewer #1: Thank you for your revision. You have addressed my questions and comments and I am satisfied with your revision.

7. PLOS authors have the option to publish the peer review history of their article (what does this mean?). If published, this will include your full peer review and any attached files.

Reviewer #1: **Yes: **Saeed Ghorbani

---

## [Editor Report · Acceptance letter]

23 Nov 2022

PONE-D-22-24485R1 

An external focus of attention enhances table tennis backhand stroke accuracy in low-skilled players 

Dear Dr. Niźnikowski:

I'm pleased to inform you that your manuscript has been deemed suitable for publication in PLOS ONE. Congratulations! Your manuscript is now with our production department. 

Kind regards, 

on behalf of

Dr. Danica Janicijevic 

Academic Editor

PLOS ONE